# Growth Plate Pathology in the Mucopolysaccharidosis Type VI Rat Model—An Experimental and Computational Approach

**DOI:** 10.3390/diagnostics10060360

**Published:** 2020-05-31

**Authors:** Johana M. Guevara-Morales, Michael Frohbergh, Hector Castro-Abril, Juan J. Vaca-González, Luis A. Barrera, Diego A. Garzón-Alvarado, Edward Schuchman, Calogera Simonaro

**Affiliations:** 1Institute for the Study of Inborn Errors of Metabolism, (IEIM), Pontificia Universidad, Javeriana, Bogotá 4665684, Colombia; 2Exponent Inc., Philadelphia, PA 19104, USA; mfrohbergh@exponent.com; 3Numerical Methods and Modeling Research Group (GNUM)-Biomimetics Laboratory, Universidad Nacional de Colombia, Bogotá 4665684, Colombia; hacastroa@unal.edu.co (H.C.-A.); dagarzona@unal.edu.co (D.A.G.-A.); 4Nefertiti, Wellness and New Technologies Group, School of Health and Sports Sciences, Fundación Universitaria del Área Andina, Bogotá 4665684, Colombia; jvaca8@areandina.edu.co; 5Institute for the Study of Inborn Errors of Metabolism, (IEIM), Pontificia Universidad, Javeriana-Clinic for Inborn Errors of Metabolism (CEIM)-Hospital Universitario San Ignacio, Bogotá 4665684, Colombia; abarrera@javeriana.edu.co; 6Department of Genetics and Genomic Sciences, Icahn School of Medicine at Mount Sinai, New York, NY 10029, USA; edward.schuchman@mssm.edu (E.S.); calogera.simonaro@mssm.edu (C.S.)

**Keywords:** mucopolysaccharidosis type VI, growth plate histology, columnar arrangement, skeletal dysplasia, growth plate computational model

## Abstract

Background: Mucopolysaccharidoses (MPS) are a group of inherited metabolic diseases caused by impaired function or absence of lysosomal enzymes involved in degradation of glycosaminoglycans. Clinically, MPS are skeletal dysplasias, characterized by cartilage abnormalities and disturbances in the process of endochondral ossification. Histologic abnormalities of growth cartilage have been reported at advanced stages of the disease, but information regarding growth plate pathology progression either in humans or in animal models, as well as its pathophysiology, is limited. Methods: Histological analyses of distal femur growth plates of wild type (WT) and mucopolysaccharidosis type VI (MPS VI) rats at different stages of development were performed, including quantitative data. Experimental findings were then analyzed in a theoretical scenario. Results: Histological evaluation showed a progressive loss of histological architecture within the growth plate. Furthermore, in silico simulation suggest the abnormal cell distribution in the tissue may lead to alterations in biochemical gradients, which may be one of the factors contributing to the growth plate abnormalities observed, highlighting aspects that must be the focus of future experimental works. Conclusion: The results presented shed some light on the progression of growth plate alterations observed in MPS VI and evidence the potentiality of combined theoretical and experimental approaches to better understand pathological scenarios, which is a necessary step to improve the search for novel therapeutic approaches.

## 1. Introduction

Mucopolysaccharidoses (MPS) are a group of inherited metabolic disorders caused by mutations in the genes that codify for lysosomal enzymes involved in the intracellular catabolism of glycosaminoglycans (GAGs), important constituents of the extracellular matrix (ECM) of different tissues, leading to their accumulation [1,2,3].

Skeletal pathology in MPS has been mainly related to cartilage abnormalities and disturbances in the process of endochondral ossification [1,2,3]. Such s process is responsible for long bone development during the embryological period, as well as long bone growth, which is controlled by a cartilaginous structure called the growth plate, located in the metaphysis of long bones.

Histologically, the growth plate is characterized by four distinct zones: resting, proliferative, hypertrophic, and calcification [4,5,6,7,8]. In the resting zone, chondrocytes display a round morphology and the tissue is rich in the extracellular matrix. In the proliferative zone, chondrocytes acquire a flattened morphology and display a characteristic columnar arrangement parallel to the growth axis [4,5,6,7,8]. The hypertrophic zone is composed by chondrocytes that undergo terminal differentiation characterized by cell cycle arrest and cell volume increase. These hypertrophic chondrocytes are responsible for the synthesis of proteins that promote extracellular matrix calcification and vascular and osteoblast invasion. Finally, the calcification zone is composed of apoptotic cells and evidence of calcification of the matrix [4,5,6,7,8].

Structural abnormalities within the growth plate of MPS-affected individuals include severe alterations in chondrocytes morphology and organization, such as resting zone chondrocytes enlargement, low proliferation rates, and loss of columnar arrangement [3,9,10,11,12,13,14,15]. Moreover, accumulation of GAG-rich material in the extracellular matrix (ECM) of the tissue, as well as disturbances in other major components, such as collagen type II, have been observed [15,16]. However, most of these alterations have been reported at advanced stages of the disease, and there is little information available regarding the progression of growth plate pathology during early development either in humans or in the MPS animal models. In addition, specific aspects of the physiopathological mechanisms associated with those abnormalities and their temporal evolution have been poorly explored in these diseases [3,11].

To better understand skeletal compromise in MPS, we used as a model mucopolysaccharidosis type VI (MPS VI) or Maroteaux–Lamy syndrome, an MPS subtype caused by the deficiency of *N*-acetylgalactosamine 4-sulfatase (or Arylsulfatase B), a lysosomal enzyme involved in dermatan sulfate (DS) and chondroitin 4 sulfate (C4S) catabolism [1,17]. In humans, MPS VI is characterized by a severe skeletal dysplasia compromising both axial and appendicular skeleton in which abnormalities have been mainly attributed to abnormalities in endochondral ossification, although macrocephaly and craniosynostosis observed in patients may also be due to altered intramembranous ossification [3,18]. In addition, patients display visceromegaly, cardiac involvement, corneal opacity, and normal cognitive function [1,17,18]. Different animal models have been described including cats, rats, and dogs, in which the disease occurred naturally, up to the knock out mouse model generated in a laboratory [19]. Here, we used the rat model of MPS VI, since it reproduces the bone pathology observed in humans. In fact, the affected animals present with smaller bodies, shorter limbs than normal, and display facial dysmorphia, as well as severe histological abnormalities of growth plate and articular cartilage [15,20,21].

Taking the above-mentioned into account, and in order to further understand growth plate pathology in MPS and its time evolution, we describe a characterization of the growth plate compromise in a rat model of MPS VI at three different stages of development (4-days-old, 1-months-old, and 3-months-old). Considering that growth plate structure imposes specific constrains on biochemical gradients and cell growth in the tissue, we then hypothesize that histological alterations observed may have an impact on the mechanical and biochemical behavior of the tissue. To make an initial exploration of such a hypothesis, experimental findings were analyzed in a theoretical scenario to better understand their potential impact on growth plate function and, consequently, in bone growth. Our results showed that, although at birth MPS VI growth plate chondrocytes organization was not altered, in older animals, chondrocyte columnar arrangement and hypertrophy is altered in a progressive way. Furthermore, in silico simulation of the observed column misalignment reveals that abnormal cell distribution in the tissue may lead to alterations in biochemical gradients that may be one of the factors contributing to the general loss of architecture and short stature observed in both the animal model and human pathology. These results shed some light on the pathophysiological mechanisms associated with bone alterations observed in MPS VI and evidence the potentiality of combined theoretical and experimental approaches to better understand pathological scenarios, which is a necessary step to improve the search for novel therapeutic approaches.

## 2. Materials and Methods

### 2.1. Characterization of Growth Plate Pathology in the MPS VI Rat Model

#### 2.1.1. Animals

The MPS VI rat model displays biochemical, pathological, and phenotypic manifestations that resemble the human phenotype. Abnormal findings include: severe skeletal compromise characterized by short limbs, growth retardation, and fascial dysmorphia that become apparent beyond 3 weeks of age; deficient arylsulfatase B enzyme activity; GAG accumulation in urine and tissues; and microscopic alteration of cartilaginous tissue architecture, as well as macrophages, cardiac valve fibroblasts, cornea, vascular smooth muscle cells, and uterine stromal cells [15,21,22].

Age-matched wild type and MPS VI Sprague–Dawley rats were used for histological analysis. Euthanasia of rats was performed using carbon dioxide inhalation. All animal protocols were approved by the Mount Sinai Institutional Animal Care and Use Committee (protocol # 08-0108, 7 April 2011) and were performed in accordance with National Institute of Health (NIH) guidelines.

#### 2.1.2. Sample Collection

In order to analyze bone samples at different ossification states corresponding to a completely cartilaginous epiphysis, an epiphysis in ossification, and a completely ossified epiphysis, distal femurs were isolated from 4-day, 1-month, and 3-month-old rats (*n* = 3/group), respectively. Samples were fixed in buffered 10% formalin.

#### 2.1.3. Histological Analyses

Samples were serially dehydrated through ethanol/xylene and embedded in paraffin. Prior to dehydration and embedding, 1- and 3-month-old samples were decalcified in Cal-Ex (Thermo Fisher Scientific, Waltham, MA, USA) for 3–7 days. Embedded samples were sectioned at 5 µm thickness, obtaining coronal sections of the femur. Slides were rehydrated through xylene/ethanol/water and stained with Harris modified hematoxylin and eosin Y (H&E, Sigma Aldrich, St. Louis, MO, USA) and Toluidine Blue (TB, Sigma Aldrich, St. Louis, MO, USA), following standard protocols.

In addition, immunostaining was performed using the Ultravision Detection System (Lab Vision Corporation, Fremont, CA, USA, TR-15-HD) to detect the Col II (Santa Cruz Biotechnology, Dallas, TX, USA, sc-28887) antibody signals. Briefly, sectioned slides were baked at 65 °C for 1 h, deparaffinized, and rehydrated through serial xylene/ethanol gradients to deionized water. Antigen retrieval was performed using 1% proteinase K for 15 min at 37 °C. Slides were blocked using hydrogen peroxide and the specific kit components. They were then incubated with anti-Col II antibody solution, containing 5% serum and 0.1% Tween-20 at 4 °C overnight (at 1:250 dilution). The following day, the slides were brought to room temperature (RT) and incubated with secondary antibody solution for 30 min at RT (biotinylated goat anti-rabbit). The slides were then incubated in streptavidin peroxidase for 10 min and imaged with 3’-diaminobenzidine (DAB) chromogen until the tissues turned brown. Hematoxylin was used as a counterstain.

The staining area was analyzed using ImageJ, following the protocol published in [23] on 20×-magnified images by using at least 2 images per individual (1 image × 2 independent sections). A total of 3 animals were included per group.

#### 2.1.4. Quantitative Analysis of Growth Plates

Three histomorphometric parameters (total growth plate (GP) thickness, resting and proliferative zone (RPZ) thickness, and hypertrophic zone (HZ) thickness) were measured to describe growth plates in wild type and MPS VI animals (Figure 1A). Growth plate zones were defined based on cell size, as described by Valteau et al. [24]. For quantification, a modified method from Valteau et al. [24] was applied. Briefly, measurements were performed on 10× magnified images, according to a minimum of 90 measures that were obtained for each group through evaluation of at least 3 images per individual (1 image × 3 independent sections). Ten measures were performed per image, and a total of 3 animals were included per group.

In addition, a quantitative description of growth plate columnar organization in proliferative, pre-hypertrophic, and hypertrophic zones in wild type and MPS VI animals was performed, as previously described [25]. Grid-based cell quantifications of the zones were performed on 20× magnified images (Figure 1B). At least 3 images per individual (1 image × 3 independent sections) were analyzed, in which four parameters were taken into account: cellular density (C), defined as the total number of cells within a grid field (cells are included if more than 90% of the area is within the field); column density (CD), corresponding to the total number of columns within a grid field; density of isolated cells (CI), defined as the number of cells within a field that do not belong to a column; and column orientation (α), which represents the angle formed between the line that connects the geometrical centers of each cell within the column (which is the column axis) and the reference axis corresponding to the ossification front (Figure 1C) [25]. Since the parameter CI illustrates the quality of columnar arrangement within the growth plate, the results obtained for this parameter are reported as the proportion of isolated cells among the total cell within a field (CI/C). All measurements were performed by image analysis using the ImageJ Java source code tool in a blind manner. Results per zone were averaged and expressed as cell per area (mm^2^).

#### 2.1.5. Statistical Analysis

Results are presented as mean ± standard error mean. To determine levels of significance of the differences between wild type and MPS VI, results were analyzed using the Mann-Whitney U test. Differences were considered significant at *p* < 0.05. Graph Pad Prism Version 3.1 (Graph Pad Software, San Diego, CA, USA) was used for statistical analysis.

### 2.2. In Silico Analysis

Cellular alterations observed in MPS VI growth plates were simulated using a computational mechanobiological model for cell hypertrophy, previously developed using the finite elements method [26]. Briefly, a bi-dimensional domain representing a segment of a growth plate column was used. The domain included eight chondrocytes between two cartilage zones located at the top and the bottom. The model simplified tissue composition in three specific regions: cells, pericellular matrix, and extracellular matrix, attending to their different composition and mechanical behavior (Figure 2A). For all structures, mechanical behavior was considered as linear elastic and mechanical properties for each tissue were taken from Castro-Abril et al. 2017 [26].

The model simulates the transition from the proliferative to hypertrophic state of the cells, considering that the process is mainly regulated by biochemical factors, mechanical strains, and time. Thus, the biochemical component is given by two factors: Indian hedgehog protein (Ihh) and parathyroid hormone-related protein (PTHrP). In vivo, these molecules interact, forming a regulatory feedback loop such that Ihh is a morphogen synthetized by chondrocytes on the first stages of hypertrophy. It diffuses through the growth plate, stimulating PTHrP synthesis by resting chondrocytes [4,8,27,28,29,30,31]. In turn, PTHrP inhibits Ihh synthesis indirectly by delaying hypertrophy and maintaining cells in a proliferative state [27,28,29,30,31]. Based on this mechanism, the Ihh-PTHrP regulatory loop was modelled as a reaction-diffusion equation system considering that Ihh is produced by cells undergoing first stages of hypertrophy while PTHrP is produced in the upper boundary of the model where the resting zone would be located (Figure 2B). PTHrP and Ihh production constants were established by numerical estimation, such that a physiological behavior could be represented due to the lack of experimental evidence (for further details see Reference [26]). In addition, concentration changes in time of both molecules is given by their diffusion through the extracellular matrix and half-life; values that were established based on literature reports [26]. Considering that the model represents a segment of the growth plate surrounded by similar units and that the biochemical effect is local, no flux of molecules was considered through the boundaries.

Cell hypertrophy is triggered by the drop of PTHrP values below the estimated threshold on the cell periphery. Once the process initiates cell growth, it depends on time and strains, thus, a function of cell growth was established based on experimental findings, leading to progressive cell growth in a timeframe of 24 h. In this time, cell deformation is affected by the hydrostatic stresses sensed by each cell, according to the Hueter-Volkman law that states that compressive stresses inhibit growth while tensile stress promotes it (complete mathematical formulation is detailed in [26]).

To start the simulation, an initial Ihh concentration was used as the input and a complete timeframe of the simulation corresponded to 250 time steps (approximately 250 h of real time) to follow the hypertrophy process of at least three cells at a reasonable computational cost. Solving was performed using finite element analysis for spatial discretization and a backward Euler for temporal discretization. Numerical solutions were calculated using a Fortran user developed subroutine.

For the purposes of this work, two computational scenarios were simulated. The physiological model consisted of a column of cells perfectly aligned with the growth axis. On the other hand, a second geometric model resembling the experimental findings in MPS VI animals was generated by inclining the cell column by 30°, with respect to the growth axis, while maintaining the size for PCM (Figure 2A). Such inclination corresponds to the maximum deviation observed in MPS growth plate columns orientation, (see Table 2 for details). The model also included a wider ECM zone. Material properties, constrains, and formulation of biochemical parameters within the model were previously described [26]. Simulations were performed without mechanical loading.

## 3. Results

### 3.1. Histological Description

At 4 days of age, MPS VI animals did not display any abnormalities either in terms of growth plate zones size or columnar arrangement (Table 1, Figure 3).

Similar to the 4-day-old animals, no statistically significant differences were observed in the thickness of growth plates of 1-month-old animals (Table 1). However, disturbances in the zonal arrangement were evidenced, consisting of a decrease in hypertrophic zone thickness (Table 1). Furthermore, quantitative analysis of columns within the growth plate revealed alterations in chondrocytes distribution along the different zones of the growth plate in MPS VI animals compared to wild type (Figure 4). Such abnormalities involved the proliferative zone, where a reduced number of chondrocytes was found in MPS VI samples (Figure 4B). In these samples, a lower tendency to arrange in columns was evidenced by a decreased columnar density and an increased rate of isolated cells (Figure 4C,D). In contrast, increased cell density was observed in pre-hypertrophic and hypertrophic zones of MPS VI animals (Figure 4A). In terms of columnar orientation, although the mean angle of columns between wild type and MPS VI animals were very similar in all zones, it was observed that in MPS VI animals the orientation of chondrocytes columns was more heterogeneous, as evidenced by an increase in the variation coefficient (Table 2).

The qualitative observation of 1-month-old bones showed a broader zone of cartilage towards the epiphyseal side of the plate (yellow arrows in Figure 5A,B), as well as in the articular surface of the epiphysis (black arrows in Figure 5B), which was observed in MPS VI animals. Last, cell size seemed to be diminished in MPS VI growth plates (Figure 4A).

Finally, 3-month-old MPS VI animals showed no alterations either in growth plate thickness (Table 1). However, complete loss of zonal arrangement in the internal structure of the growth plate was observed (Figure 6A). In consequence, quantitative analysis for these samples was performed using a 50 × 50 µm grid for all the growth plates for normal and MPS VI samples of this age. Such analyses revealed a decrease in both cell and columnar density (Figure 6B,C). Moreover, a high proportion of cells were not organized in columns, as evidenced by the increase in the isolated cell ratio (Figure 6D). Similar to the observed for 1-month-old samples, a higher variation in column angles was observed in MPS VI growth plates (Table 2).

Major morphological characteristics of wild type (WT) and MPSVI growth plates were analyzed by qualitative observation of a small magnification of sections stained with haematoxylin-eosin. At 4 days of age, MPS VI animals did not display evident abnormalities in the epiphyses shape. Moreover, a region of hypertrophy in the central portion of epiphyses suggestive of secondary ossification center development was observed, and likewise in both populations (yellow arrows in Figure 7). In terms of the ossification front, it was observed that MPS VI bones displayed a subtle depression in the central part, acquiring an “m” shape, while they were completely flat in WT bones (red arrow in Figure 7); this change in shape of the growth plate was even more evident in 1-month-old animals. In contrast, no differences were observed in 3-month-old MPS VI animals (Figure 7).

To further characterize growth plate abnormalities in older MPS VI animals, the two main components of the growth plate extracellular matrix were analyzed: GAG (by toluidine blue staining (TB)) and collagen type II (Col II) (by immunohistochemistry). Immunostaining for col II showed a stronger staining tendency in wild type animals compared to MPS VI at 1 month and 3 months of age, although they were not statistically significant (Figure 8). In contrast, for toluidine blue staining, only 3-month-old MPS VI animals showed differences with the WT group. In these animals, the strong blue staining indicates GAG accumulation in these animals (Figure 8). No differences were observed at 4 days of age.

### 3.2. In Silico Analyses

To analyze the effect of columnar misalignment on cellular behavior, experimental findings were analyzed using a mechanobiological model for cell hypertrophy. The model considered gradients of the two main molecular regulators of this process, parathyroid hormone-related protein (PTHrP) and Indian hedgehog protein (Ihh), as well as the time evolution of the increase in cellular size characteristic of this phenotypic transition (for details see Section 2.2). Results showed that only by altering cell distribution within the domain, gradients of PTHrP and Ihh were severely disturbed, causing delay in the hypertrophy process and, subsequently, the growth of the structure. Furthermore, asymmetric concentration distribution patterns of both biochemical species were also present (Figure 9). Consequently, after 250 h of simulation, two out of eight cells underwent hypertrophy with an increase of the cell column of 25% in the misaligned model. In contrast, in a column perfectly aligned with the growth axis, four out of eight cells completed the hypertrophy process, leading to a 39% increase in column height (Figure 9). In addition, cells in the misaligned model presented changes in their shape, with respect to the WT model. Indeed, the major axes of all cells within the column tilted, with respect to the longitudinal axis of the bone with cells from the lower part of the column, presenting higher deformation, while cells in the upper part tended to be more flattened. This phenomenon lead to an asymmetric distribution of the surrounding ECM, as seen in Figure 9.

## 4. Discussion

Mucopolysaccharidoses (MPS) are genetic diseases that severely compromise hyaline cartilage structure by altering tissue composition due to the accumulation of partially degraded glycosaminoglycans (GAGs) [1,3]. These abnormalities have proven to affect not only adult hyaline cartilage, but also transient cartilage located at the growth plate, disturbing the process of endochondral ossification [3]. Although great effort has been made toward describing articular cartilage pathology and skeletal abnormalities in MPS, as well as developing novel molecular therapies, little is still known regarding the characteristics and basic pathophysiological mechanisms associated with growth plate involvement in MPS. Thus, in this context, this work gives a detailed description of growth plate pathology observed in MPS using an animal model and provides information regarding the progression of the growth plate abnormalities in MPS VI rats. Furthermore, this work aims to show a combined experimental and theoretical approximation to pathophysiological mechanisms taking place within growth plates in these diseases from a mechanical and structural point of view, using a combined computational and experimental approach.

To better understand skeletal compromise in MPS, we characterize the growth plate pathology and its progression in the rat model of MPS VI. The rat model of MPS VI reproduces the bone pathology observed in humans. In fact, the skeletal phenotype of affected animals becomes evident between 1 and 2 months of age [15]. Based on this information, the time points studied here corresponded to a neonatal stage (day 4); the second time point was selected around the onset of clinical signs and before sexual maturity (1 month). Lastly, growth plates of young adult animals (3 month) were analyzed, since at this age the animals display all phenotypical abnormalities characteristic of the disease in rats.

Results from the histological evaluation were comparable to the observed phenotype in humans, where MPS VI is characteristically a progressive disease. In fact, MPS VI patients are usually asymptomatic during the first year of life, and skeletal abnormalities become evident during early childhood [17,32]. In a similar way, no histological changes were observed in newborn MPS VI rats (4 days old) (Table 1, Figure 3). Furthermore, structural abnormalities in these animals started to become evident by 1 month of age, involving all growth plate zones: proliferative, pre-hypertrophic, and hypertrophic. Abnormalities observed were primarily related to cell density rather than columnar arrangement (Figure 4). Cellular abnormalities identified in 1-month-old MPS VI growth plates seem to indicate a rapid transition from proliferative to hypertrophic zones, since cell density decreased in the former and increased in the latter. Moreover, most of the cells within MPS VI growth plates were found in the pre-hypertrophic zone, suggesting that although more cells started hypertrophy, this process may be slower (Figure 4B). Such a hypothesis is consistent with the observed retardation in epiphyseal ossification, evidenced by the wider hypertrophic cartilage present surrounding the secondary ossification center in 1-month-old MPS VI animals (Figure 5). A similar proliferative zone involvement has been observed in the MPS VII mouse model, which shares dermatan and chondroitin-4-sulphate accumulation with MPS VI. Furthermore, in this model, loss of proliferative chondrocytes was associated with a decrease in cell proliferation, which may be an additional mechanism involved in the alterations observed in MPS VI rats that was not addressed in this study [11,33]. In the MPS VII dog and mouse model, retardation in the development of secondary ossification centers was observed for vertebral bodies and long bones. In fact, in this model, they have also reported delayed hypertrophy within the growth plates, although no histological abnormalities were observed [34]. In addition, previous studies have reported reduced metaphyseal trabecular density in the MPS VI rat model [35]. Such findings might be related to the altered hypertrophy process; however, it was not assessed in this study.

Contrasting to the cell density increase observed in the hypertrophic zone of 1-month-old MPS VI rats, the thickness of this zone was reduced (Table 1). These results suggest a decrease in cell size in affected animals that, although not measured, was qualitatively evident in microscopic images (Figure 4A). Such findings differ from the observations made in the MPS VI cat model, in which higher and vacuolated cells were identified in hypertrophic zones, although a similar compromise of the proliferative zone was described [13]. These differences may be related to changes in GAG composition of cartilage among different species, leading to different degrees of accumulation and cell involvement [36,37]. However, further histological studies are required to analyze the progression of intracellular storage in growth plate chondrocytes of MPS VI animals, since, in this study, we only analyzed samples from young animals (up to 3 months).

In addition to the cellular alterations described, morphological evolution of the growth plate revealed differences in the shape of the ossification front between WT and MPS VI animals, mainly at early stages prior to complete ossification of the epiphysis (Figure 7). These changes may be a factor altering mechanical environment in MPS VI growth plates, considering results published previously by our group and other groups [38,39]. Moreover, mechanical alterations of the tissue may be potentiated as a consequence of the biochemical disturbances observed in major components of the cartilage matrix, GAG, and collagen type II (Figure 8), which are consistent with previous reports from articular tissues of MPS patients and animal models [16,40]. Based on the genetic defect of MPS VI, accumulation of GAGs in the extracellular environment (ECM and pericellular matrix (PCM)) is expected [3,11,13,41,42,43]. Furthermore, alterations observed in collagen type II (Figure 8) might be related to the fact that dermatan sulfate (DS) accumulation in MPS VI may alter ECM and PCM structure, since DS containing proteoglycans has been found in association with collagen type II fibers and seems to be implicated in regulating fibrilogenesis through interactions mediated by DS chains [44,45,46,47,48,49,50].

Considering that collagen may be involved in maintaining columnar arrangement within the growth plate by forming fibrils orientated parallel to the growth axis in proliferative and hypertrophic zones [51,52,53,54,55,56], and GAGs are responsible for providing cartilage compression resistance [45,46,57], structural abnormalities observed in MPS VI animals may have implications in the mechanical environment within the tissue. In fact, alterations in the mechanical properties of the cartilaginous tissue have been reported in the canine model of MPS VII and the rat model of MPS VI at later stages (adult animals) [58,59]. These findings are interesting, since it has been evidenced that mechanical stimulation affects growth plate behavior in physiological and pathological conditions [60,61,62]. Thus, taken together, the evidence presented here and the evidence available in literature suggests that mechanical contribution to pathology development in MPS is a scenario that requires further studies.

Finally, taking into account that the loss of columnar arrangement was the most striking feature in 3-month-old MPS VI rats (Figure 6), the potential effect of columnar disorganization on growth plate function was assessed using an in silico mechanobiological model. Therefore, a model where chondrocytes were not aligned to the bone longitudinal growth axis was considered. Our results support the idea that columnar orientations may be an important factor affecting normal hypertrophy, since it was observed that only by modifying cell arrangement, there was a disturbance in the biochemical environment that severely delayed cell hypertrophy in the model (Figure 9). This correlates to the initial abnormalities observed in 1-month-old animals that, although did not lose columnar arrangement, displayed a higher variability in columnar orientation and an important delay on chondrocyte hypertrophy, as discussed above. Moreover, it is interesting that, although no differences were observed in toluidine blue staining (GAG accumulation) of the extracellular matrix, alterations were evident in terms of cell morphology in MPS animals, presenting with bigger and vacuolated cells and a tendency to lighter staining of Col II (Figure 8). Such results may be related to the fact that lysosomal storage disorders imply slow progressive accumulation of substrates that is initially intracellular and then affects the extracellular medium, first due to altered cell functions, such as secretion of extracellular components, and later on by secretion of accumulated material, which coincides with histological findings at 3 months (Figure 8) [63]. Based on these observations, we hypothesize that primary biochemical abnormality in MPS VI leads to changes in extracellular matrix structure that promotes chondrocytes disorganization within the growth plate. As a consequence, normal biochemical gradients responsible for growth plate normal regulation are affected leading to complete disruption of the normal endochondral ossification process. Such a scenario may be possible considering the proposed role of DS chains in collagen nets assembly. Furthermore, appropriate ECM structure has been suggested as one of the factors involved in promoting and maintaining columnar organization, as evidenced by animal models. In fact, preservation of columnar arrangement until 1 month may explain why these animals did not display evident skeletal abnormalities at that age despite the other histological alterations identified (Figure 3; Figure 4).

It is important to note that the approximation presented here has some limitations, since the evidence presented is circumscribed to distal femur epiphysis. Moreover, important histological changes were observed between the time points analyzed, suggesting that studying intermediate points and other growth plate localizations might be useful for better understanding growth plate pathology in MPS VI. On the other hand, the computational analyses presented are based on a theoretical scenario that implies several assumptions, mainly around the lack of experimental data. In this sense, although the model used was already proved to reproduce a physiological behavior, we simulated only structural abnormalities, assuming a normal biochemical behavior in MPS VI growth plates. In addition, due to computational cost, simulation was limited to a unique column in 250 steps and takes into account only a select set of biochemical and mechanical factors. Therefore, it is difficult to extrapolate these findings to the in vivo context. However, our approximation constitutes a proof of concept that mechanobiological models can be used to study mechanical and biochemical interactions taking place in biological processes. In fact, the formulation of this model lead to the identification of some knowledge gaps that need to be fulfilled in order to improve these kinds of models, and those aspects must be the focus of future works. These include more detailed information regarding: (1) Ihh/PTHrP behavior in MPS VI; (2) mechanical properties of MPS VI tissue; and (3) mechanical environment within MPS VI growth plates, among others. Moreover, computational results suggest potential physiopathological mechanisms that deserve further experimental validation, highlighting the potential of such models for direct experimental efforts.

## 5. Conclusions

In conclusion, our experimental data suggest that the main consequence of GAG accumulation on the growth plate is chondrocytes disorganization, probably associated with an abnormal ECM structure. The theoretical evidence presented here provides additional information about the potential mechanisms involved in the disease, showing that column disorganization may lead to an abnormal biochemical environment within the plate. Thus, our data suggest that loss of columnar arrangement is an early event that may trigger loss of growth plate normal function, although it might not be the only one. In fact, alterations in intracellular signaling pathways involving pathways related to signal transducer and activator proteins (STATs) and Janus kinases (JAKs) have been described as involved in growth plate pathology in MPS VII mice [11]. Moreover, alteration in bone morphogenic proteins (BMPs) and transforming growth factor (TGFs) signaling, autophagy, and inflammation have been described as playing a role in cellular dysfunction in lysosomal storage diseases, although their role in growth plate pathology in MPS has been poorly studied [3,63,64]. Therefore, although our results require further experimental confirmation, the combined experimental and computational approached used here allowed the definition of a new research path in order to better understand the growth plate pathology in MPS that can be applied to other genetic chondrodysplasias. These kinds of studies contribute greatly to elucidating pathological mechanisms in these diseases and may expand the understanding of bone pathology, a necessary step in the development of novel therapeutic approaches.

## Figures and Tables

**Figure 1 diagnostics-10-00360-f001:**
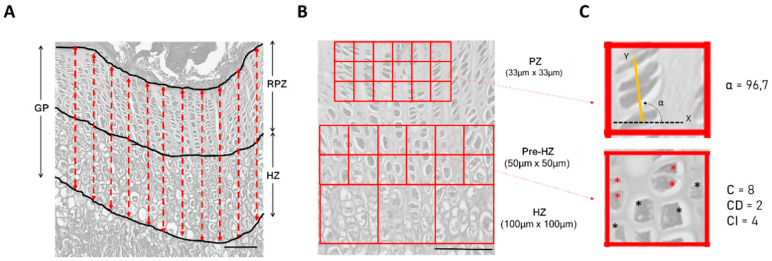
Image processing for quantitative analysis of wild type 1-month-old rat growth plates. (**A**) Evaluation of total growth plate (GP) and zonal thickness (resting andproliferative zone (RPZ); hypertrophic zone (HZ)). 10× magnification; (**B**) Grid used for evaluation of columnar organization per zone, as previously described in [25] (proliferative zone (PZ); pre-hypertrophic zone (Pre-HZ); hypertrophic zone (HZ) 20× magnification. Scale bars = 100 µm; (**C**) Illustration of quantitative description of growth plate columnar organization. The upper panel shows the calculation of α where the axis labeled as Y (yellow line) symbolizes the line that links the geometric centers of the cells that form the column, while the axis labeled as × (dotted black line) represents the transversal axis to the one in the preferential bone growth (corresponding to the ossification front). α represents the column orientation angle. The lower panel illustrates the calculation of parameters cellular density (**C**), CD, and CI. Asterisks mark the cells quantitated within the field; columnar cells are marked in red and isolated cells are marked in black.

**Figure 2 diagnostics-10-00360-f002:**
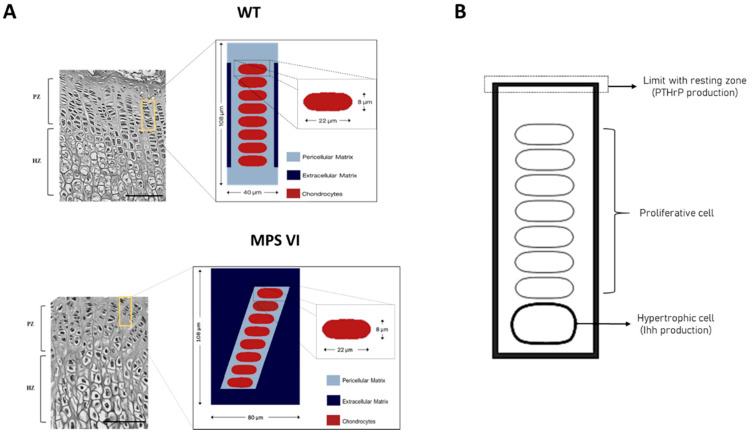
Graphical description of the work domain used in the computational mechanobiological model. (**A**) The model corresponding to wild type animals considers a column of cells perfectly aligned with the growth axis (Image modified from [26]), while for mucopolysaccharidosis type VI (MPS VI) animals it is considered misaligned as indicated in the text and corresponding to experimental findings. Scale bars correspond to 100 µm; (**B**) Localization of the structures responsible for the production of PTHrP and Ihh.

**Figure 3 diagnostics-10-00360-f003:**
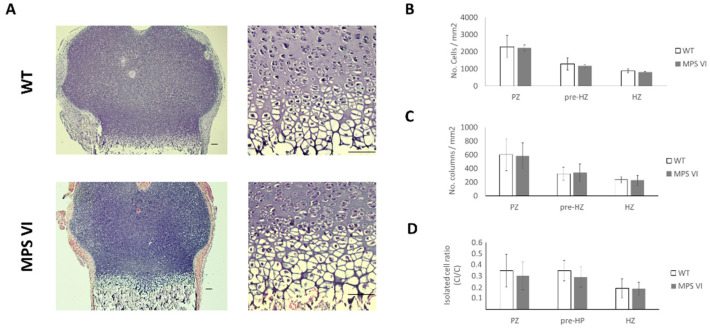
Histological characteristics of growth plates from 4-day-old wild type (WT) and mucopolysaccharidosis type VI (MPS VI) rats. (**A**) Histological sections stained with hematoxylin and eosin images at 4× (**left**) and 20× (**right**) magnification. Scale bars correspond to 100 µm; (**B**) Cell density results per zone (C); (**C**) Column density (CD) results per zone; (**D**) Results for isolated cells ratio per zone. Open and black bars correspond to wild type (WT) and MPS VI rats, respectively. Results are reported per growth plate zone as follows: proliferative (PZ), pre-hypertrophic (pre-HZ), and hypertrophic (HZ) zones (*n* = 3 per group).

**Figure 4 diagnostics-10-00360-f004:**
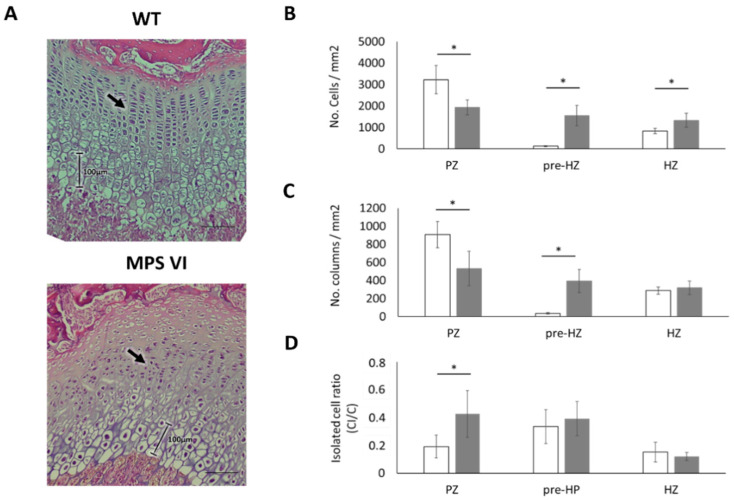
Histological characteristics of growth plates from 1-month-old WT and MPS VI rats. (**A**) Histological sections stained with hematoxylin and eosin. In the images, a severe loss of columnar arrangement is observed (arrow head). In addition, a change in the size of hypertrophic chondrocytes is observed. Images are at 20× magnification. Scale bars correspond to 100 µm; (**B**) Cell density results per zone; (**C**) Column density (CD) results per zone; (**D**) Results for isolated cells ratio (CI/C) per zone. Open and black bars correspond to wild type (WT) and MPS VI rats, respectively. Result are reported per growth plate zone as follows: proliferative (PZ), pre-hypertrophic (pre-HZ), and hypertrophic (HZ) zones (*n* = 3 per group). The symbol (*) indicates a statistically significant difference (*p* < 0.05).

**Figure 5 diagnostics-10-00360-f005:**
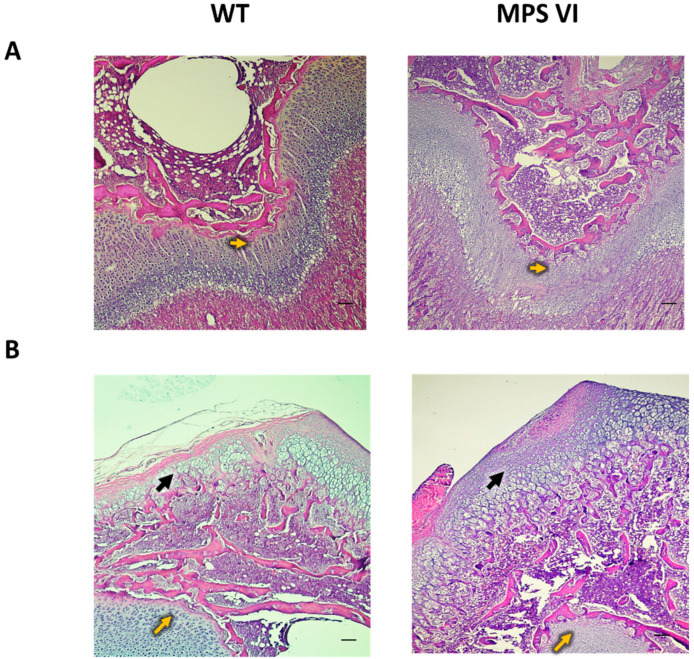
Microscopic images of distal femur epiphyses of WT and MPS VI animals. (**A**) Growth Plate; (**B**) Articular surface. Histological sections were stained with hematoxylin and eosin. Images are at 10× magnification. Scale bars correspond to 100 µm. Yellow arrows show epiphyseal side of the growth plate. Black arrows show the articular surface.

**Figure 6 diagnostics-10-00360-f006:**
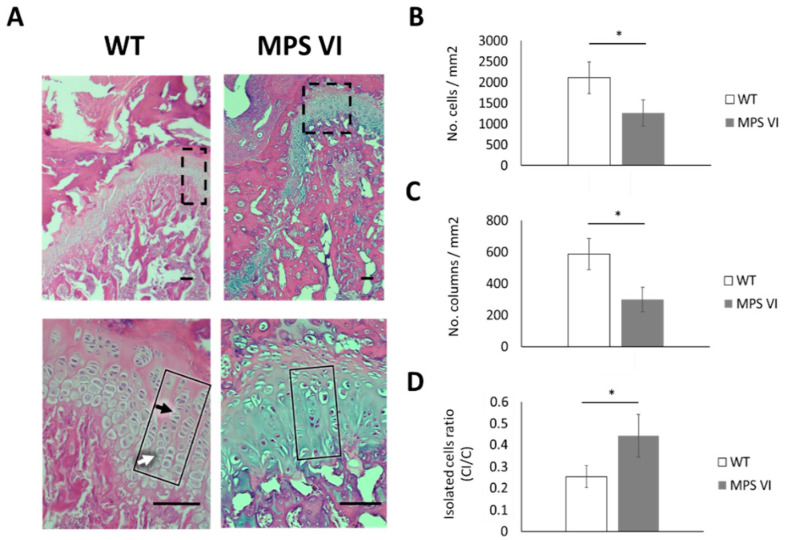
Histological characteristics of growth plates from 3-month-old WT and MPS VI rats. (**A**) The upper shows low magnification images (4×), showing no remarkable differences among WT and MPS samples. In the lower panel, higher magnification (20 v) images (corresponding to the areas highlighted in dotted lines in the upper panel) show that in the normal growth plates chondrocytes are arranged in organized columns (black rectangle) and it is possible to differentiate proliferative (black arrow) from hypertrophic (white arrow). This architecture is lost in the mucopolysaccharidoses (MPS) rats (black rectangle). Histological sections were stained with hematoxylin and eosin. Scale bars = 100 µm; (**B**–**D**) Results correspond to whole growth plate findings due to loss of zonal arrangement (see text). (**B**) Cell density; (**C**) Column density (CD); (**D**) Results for the isolated cells ratio (CI/C). Open and black bars correspond to wild type (WT) and MPS VI rats, respectively. Results are reported per growth plate zone as follows: proliferative (PZ), pre-hypertrophic (pre-HZ), and hypertrophic (HZ) zones (*n* = 3 per group). The symbol (*) indicates a statistically significant difference (*p* < 0.05).

**Figure 7 diagnostics-10-00360-f007:**
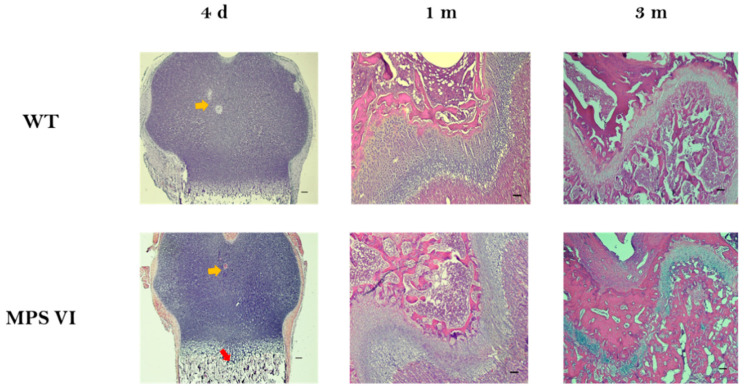
Histological characteristics of growth plates from WT and MPS VI rats. Histological sections stained with hematoxylin and eosin. Images are at 10× magnification. Scale bars correspond to 100 µm. Yellow arrow shows initial regions of ossification in the center of the epiphyses. The red arrow shows changes in the ossification front. Representative samples (*n* = 3 per group).

**Figure 8 diagnostics-10-00360-f008:**
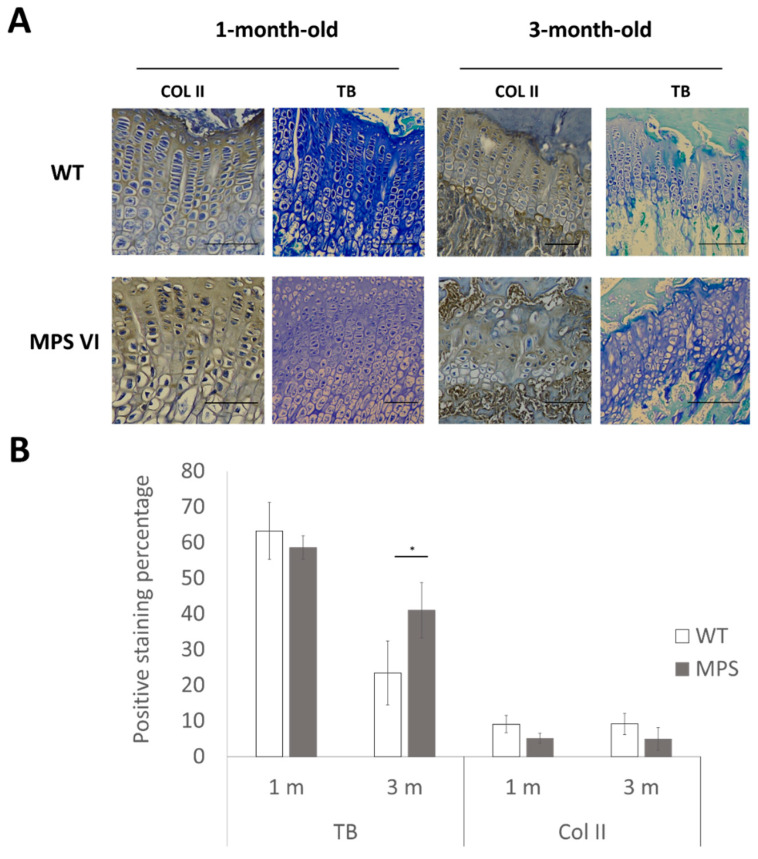
Histology-based analysis of extracellular matrix composition in wild type and MPS VI rats. (**A**) On the left side of both panels, immunohistochemistry for collagen type II (Col II) is shown. On the right side, images of toluidine blue stained (TB) samples are shown. Scale bars correspond to 100 µm. Representative samples (*n* = 3 per group); (**B**) Quantification analysis performed using Image J software (mean ± standard deviation; *n* = 3 per group). Significant differences are noted with an asterisk for *p* < 0.01.

**Figure 9 diagnostics-10-00360-f009:**
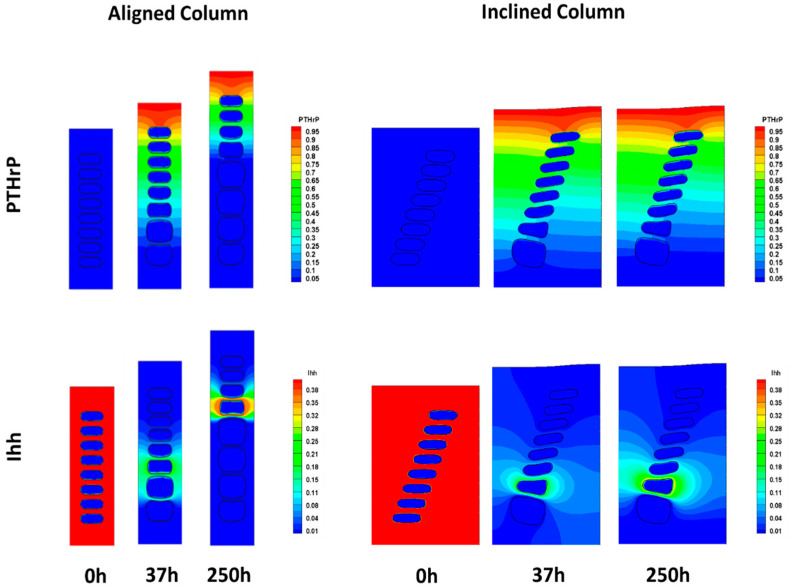
Indian hedgehog protein (Ihh) and parathyroid hormone-related protein (PTHrP) concentration changes over time for a normal and an inclined growth plate column. Images show the diffusion patterns observed for PTHrP (upper panel) and Ihh (lower panel) at times 0, 37, and 250 h of simulation using the mechanobiological model described in [26]. Results correspond to the simulations performed using morphologies corresponding to a growth plate column aligned with the longitudinal bone growth axis (**left**) and an inclined column (**right**).

**Table 1 diagnostics-10-00360-t001:** Histomorphological measurements of wild type and mucopolysaccharidosis type VI (MPS VI) growth plates.

Age	Parameter	Wild Type	MPS VI
4 days	Growth plate thickness (µm)	659.04 ± 96.28	539.5 ± 136.26
	Resting-proliferative zone thickness (µm)	461.83 ± 87.24	319.15 ± 118.35
	Hypertrophic zone thickness (µm)	197.11 ± 29.74	220.35 ± 33.02
1 month	Growth plate thickness (µm)	466.62 ± 46.16	381.68 ± 81.23
	Resting-proliferative zone thickness (µm)	225.17 ± 34.63	241.24 ± 59.75
	Hypertrophic zone thickness (µm)	241.11 ± 31.61	140.43 ± 41.74 *
3 months	Growth plate thickness (µm)	224.49 ± 33.98	244 ± 78.4

* Statistically significant difference when compared to wild type (WT) (*p* < 0.05).

**Table 2 diagnostics-10-00360-t002:** Column orientation angle and coefficient of variation for wild type and MPS VI growth plates.

Age	Growth Plate Zone	Column Orientation Angle (α) *^,§^		Variation Coefficient of α (%) *	
		Wild Type	MPS VI	Wild Type	MPS VI
4 days	Proliferative	86.77	89.38	33	28
	Pre-hypertrophic	88.53	85.99	28	39
	Hypertrophic	90.24	91.18	22	22
1 month	Proliferative	93.82	84.05	15	20
	Pre-hypertrophic	92.15	82.48	17	23
	Hypertrophic	89.4	81.51	13	21
3 months	Total growth plate	92.23	86.69	17	24

* Results presented correspond to the average value. ^§^ α represents the orientation of the column in respect to an axis perpendicular to the growth axis (see Figure 1C for details).

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
