# Peer review of "Growth Plate Pathology in the Mucopolysaccharidosis Type VI Rat Model—An Experimental and Computational Approach"

_diagnostics, 2020, doi:10.3390/diagnostics10060360_

Round 1
Reviewer 1 Report
Manuscript ID: diagnostics-655199
Type of manuscript: Article
Title: Growth plate pathology in the mucopolysaccharidosis type VI rat model. An experimental and computational approach
The authors performed histological analyses of distal femur growth plates of WT and MPS VI rats at different stages of development including quantitative data. The experimental findings were then analyzed in a theoretical scenario. This study shed some light on the progression of growth plate alterations observed in MPS VI and evidence the potentiality of combined theoretical and experimental approaches to better understand pathological scenarios, which is a necessary step to improve the search for novel therapeutic approaches.
This study is well-written and interesting. Only some minor points are needed to be clarified.
Please add the “limitation” part in the “Discussion”. Lines 113 and 138, “Hypertrophic (HZ)” should be changed to “hypertrophic (HZ)”.Author Response
The authors want to thank the reviewer for the comments. The
manuscript has been thoroughly revised; thus, we believe it has greatly improved. In the following section we present the changes included which are also highlighted within the document using the "track changes" mode.
Point 1: Please add the “limitation” part in the “Discussion”.
Response: The limitations of the study were clarifyied and broaden in the last paragraph of the discussion.
Lines 113 and 138, “Hypertrophic (HZ)” should be changed to “hypertrophic (HZ)”.
Response: According to the reviewer’s comment, we have made the suggested changes.
Reviewer 2 Report
The paper is well written and easy to follow. The authors presented progressive histological alternation of growth plates in MPS VI rat model in combination of in silico simulation to address pathophysiological mechanisms associated to bone alterations in MPS VI.
My comments include:
In abstract and introduction, the authors state “Mucopolysaccharidoses (MPS) are a group of genetic skeletal dysplasias”. Suggest to add that MPS are a group of inherited metabolic diseases caused by the absence or malfunctioning of certain lysosomal enzymes needed to break down glycosaminoglycans ….. Due to the publication format, in silico analyses is described and discussed in details in section 4.2. Some contents in section 4.2 can be moved to section 2.2. In silico analyses and section 3 for readers easy to follow the results and discussion. In section 4.1.1, it stated “The MPS VI rat model has been previously described [21, 54, 55].” Suggest to add couple sentences to describe the disease model, so the readers do not need to go to the reference publications to find information.Author Response
The authors want to thank the reviewer for the comments. The
manuscript has been thoroughly revised; thus, we believe it has greatly improved. In the following section we present the changes included which are also highlighted within the document using the "track changes" mode.
Point 1: In abstract and introduction, the authors state “Mucopolysaccharidoses (MPS) are a group of genetic skeletal dysplasias”. Suggest to add that MPS are a group of inherited metabolic diseases causribed and disced by the absence or malfunctioning of certain lysosomal enzymes needed to break down glycosaminoglycans …..
Response: According to the reviewer’s comment, we have made the suggested changes in the initial paragraph of the abstract and introduction.
Point 2: Due to the publication format, in silico analyses is descussed in details in section 4.2. Some contents in section 4.2 can be moved to section 2.2. In silico analyses and section 3 for readers easy to follow the results and discussion.
Response: Attending to the reviewr's comment, a brief introduction to the model was included in section 2.2 for better understanding the model. However the reader is referred to section 4.2 for further details, since complete explanation implies to mention several technical details that are proper of methods section rather than results.
Point 3: In section 4.1.1, it stated “The MPS VI rat model has been previously described [21, 54, 55].” Suggest to add couple sentences to describe the disease model, so the readers do not need to go to the reference publications to find information.
Response: As suggested, a brief description of the clinical and pathological findings of the animal model were included within the text.
Reviewer 3 Report
Guevara et al. show the evidence that growth plate alterations are progressive in mucopolysaccharidosis type VI rat by histopathological techniques. I have several concerns. Point 1. Page 10, line 308, authors described that “Based on these observations… that promotes chondrocytes disorganization within the growth plate”. This hypothesis is plausible. However, I feel presented data is not sufficient to support their hypothesis. Whereas histological changes are observed in growth plate from 1month-old MPS VI, Fig.6 indicated that no difference of GAG accumulation is observed between 1-month-old WT and MPS VI rat. Please explain the discrepancy. 2. In addition, histological analysis is less quantitative and not sensitive method for detection of GAG accumulation. Please add the quantitative analysis to Fig.6Author Response
The authors want to thank the reviewer for the comments. The
manuscript has been thoroughly revised; thus, we believe it has greatly improved. In the following section we present the changes included which are also highlighted within the document using the "track changes" mode.
Point 1. Page 10, line 308, authors described that “Based on these observations… that promotes chondrocytes disorganization within the growth plate”. This hypothesis is plausible. However, I feel presented data is not sufficient to support their hypothesis. Whereas histological changes are observed in growth plate from 1month-old MPS VI, Fig.6 indicated that no difference of GAG accumulation is observed between 1-month-old WT and MPS VI rat. Please explain the discrepancy.
Response: In order to clarify the point and, taking into account the suggestion o including the quantitative results, a better description of Figure 6 and higher magnification of TB staining were included. In addition, we add the following statement within the discussion section:
"lysosomal storage disorders imply slow progressive accumulation of substrates that is initially intracellular and then affects extracellular medium, first due to altered cell functions such as secretion of extracellular components and later on by secretion of accumulated material which coincides with histological findings at 3 months", as suggested by Platt et al. 2012.
Additionally, and in response to other reviewer comment, methodological limitations were included at the end of discussion section within the manuscript.
Point 2. In addition, histological analysis is less quantitative and not sensitive method for detection of GAG accumulation. Please add the quantitative analysis to Fig.6
Response: According to the reviewer’s comment, we have included in Figure 6 the results of imageJ quantification. In addittion, the methodology was included in methods section.
Round 2
Reviewer 3 Report
This manuscript was well revised by authors.